# Similarity-based Link Prediction from Modular Compression of Network Flows

**Christopher Blöcker**
Integrated Science Lab, Department of Physics
Umeå University
`christopher.blocker@umu.se`

**Jelena Smiljanić**[*]
Integrated Science Lab, Department of Physics
Umeå University
`jelena.smiljanic@umu.se`

**Ingo Scholtes**
Center for Artificial Intelligence and Data Science
University of Würzburg
`ingo.scholtes@uni-wuerzburg.de`

**Martin Rosvall**
Integrated Science Lab, Department of Physics
Umeå University
`martin.rosvall@umu.se`

## Abstract

Node similarity scores are a foundation for machine learning in graphs for clustering, node classification, anomaly detection, and link prediction with applications in biological systems, information networks, and recommender systems. Recent works on link prediction use vector space embeddings to calculate node similarities in undirected networks with good performance. Still, they have several disadvantages: limited interpretability, need for hyperparameter tuning, manual model fitting through dimensionality reduction, and poor performance from *symmetric* similarities in *directed* link prediction. We propose *MapSim*, an information-theoretic measure to assess node similarities based on modular compression of network flows. Unlike vector space embeddings, *MapSim* represents nodes in a discrete, non-metric space of communities and yields *asymmetric* similarities in an unsupervised fashion. We compare *MapSim* on a link prediction task to popular embedding-based algorithms across 47 networks and find that *MapSim*'s average performance across all networks is more than 7% higher than its closest competitor, outperforming all embedding methods in 11 of the 47 networks. Our method demonstrates the potential of compression-based approaches in graph representation learning, with promising applications in other graph learning tasks.

## 1 Introduction

Calculating similarity scores between objects is a fundamental problem in machine learning tasks, from clustering, anomaly detection, and text mining to classification and recommender systems. In Euclidean feature spaces, similarities between feature vectors are commonly calculated as lengths, norms, angles, or other geometric concepts, possibly using *kernel functions* that perform implicit non-linear mappings to high-dimensional feature spaces [1]. For relational data represented as graphs, methods using the graph topology to calculate pairwise node similarities can address learning problems such as graph clustering, node classification, and link prediction. For link prediction, recent works take a multi-step approach and separate *representation learning* and *link prediction* [2, 3]: First, they learn a latent-space node embedding from the graph's topology, using methods such as graph or matrix factorisation [4, 5], or random walk-based techniques [6–8]. Then, they interpret node positions as points in a high-dimensional feature space, possibly applying downstream dimensionality reduction. Finally, they use node positions in the resulting feature space to assign new "features" to pairs of nodes, which can be used to predict links. Taking an unsupervised approach, links are predicted based on node similarities [9] by calculating distance metrics or similarity scores between

---

[*]Also with the Center for the Study of Complex Systems, Institute of Physics, University of Belgrade.

C. Blöcker et al., Similarity-based Link Prediction from Modular Compression of Network Flows. *Proceedings of the First Learning on Graphs Conference (LoG 2022)*, PMLR 198, Virtual Event, December 9–12, 2022.

**Figure 1:** We calculate node similarities for predicting links based on a network's modular coding scheme of the map equation. Blue and orange nodes have a unique codeword within their module, shown next to the nodes and derived from their stationary visit rates. Decimal numbers show the theoretical lower limit for the codeword length in bits. *Map equation similarity*, *MapSim* for short, derives description lengths for predicted links, connecting more similar nodes uses fewer bits. Intra-community links tend to have shorter description lengths than inter-community links.

node pairs to rank them. We can alternatively use a supervised approach [10] by (i) using binary operators like the Hadamard product [7], (ii) sampling negative instances (node pairs not connected by links), and (iii) using the features of positive and negative instances to train a supervised binary classifier [7].

Advances in graph embedding and representation learning have considerably improved our ability to predict links in networks, with applications in biological [11] and social [12] networks and in recommender systems [13]. However, these methods introduce challenges for real-world link-prediction tasks: First, they require specifying hyperparameters that control aspects regarding the *scale* of patterns in graphs, the influence of local and non-local structures, and the latent space dimensionality [14]. Network-specific hyperparameter tuning addresses these issues, but is challenging in real applications and aggravates the risk of overfitting; recent systematic comparisons reveal that the performance of different methods largely varies across data sets [2, 3]. These challenges make it difficult for practitioners to choose and optimally parametrise an embedding method. Second, using latent metric spaces implies *symmetric* similarities, limiting the performance when predicting *directed* links [5, 15]. Third, compared with hand-crafted features, embeddings tend to have low interpretability: We can assess the similarity of nodes, but we cannot explain *why* some nodes are more similar than others [2–4]. Nevertheless, recent graph neural network-based approaches focus on learning features for link prediction from local subgraphs [16], overlapping node neighbourhoods [17], or shortest paths [18], achieving favourable performance. Finally, recent works highlight fundamental limitations of low-dimensional representations of complex networks [19], questioning to what extent Euclidean embeddings can capture patterns relevant to link prediction.

Motivated by recent works highlighting the importance of community structures for link prediction [2, 20, 21], we propose a novel approach to similarity-based link prediction that addresses these issues. Our contributions are:

- We introduce map equation similarity, *MapSim* for short, an information-theoretic method to calculate asymmetric node similarities. *MapSim* builds on the map equation [22], a framework that applies coding theory to compress random walks based on hierarchical cluster structures.

- Unlike other random walk-based embedding techniques, our work builds on an analytical approach to calculate the minimal expected description length of random walks, neither requiring simulating random walks nor tuning hyperparameters.

- Following the minimum description length principle, *MapSim* incorporates Occam's razor and balances explanatory power with model complexity, making dimensionality reduction superfluous. With hierarchical cluster structures, *MapSim* captures patterns at multiple scales simultaneously and combines the advantages of local and non-local similarity scores.

- We validate *MapSim* in an unsupervised, similarity-based link prediction task and compare its performance to six well-known embedding-based techniques in 47 empirical networks from different domains. This analysis highlights challenges in the generalisability of embedding techniques and parametrisations across different networks.

- Confirming recent surveys, we find that the performance of popular embedding techniques for unsupervised link prediction without network-specific hyperparameter tuning depends on the data. In contrast, *MapSim* provides high performance across a wide range of networks,

with an average performance 7.7% and 7.5% better than the best competitor in undirected and directed networks, respectively. *MapSim* outperforms the chosen baseline methods in 11 of the 47 networks with a worst-case performance 44% and 33% better than popular embedding techniques in undirected and directed networks, respectively.

In summary, we take a novel perspective on graph representation learning that fundamentally differs from other random walk-based graph embeddings. Instead of embedding nodes into a metric space, leading to symmetric similarities, we develop an unsupervised learning framework where (i) positions of nodes in a coding tree capture their representation in a non-metric latent space, and (ii) node similarities are calculated based on how well transitions between nodes are compressed by a network's hierarchical modular structure (figure 1). Apart from node similarities that can be "explained" based on community structures captured in the coding tree, *MapSim* yields asymmetric similarity scores that naturally support link prediction in directed networks. We provide a simple, non-parametric, and scalable unsupervised method with high generalisability across data sets. Our work demonstrates the power of compression-based approaches to graph representation learning, with promising applications in other graph learning tasks.

## 2    Related Work and Background

We first summarise recent works on graph embedding and similarity-based link prediction. Then, we review the map equation, an information-theoretic objective function for community detection and the theoretical foundation of our compression-based similarity score.

### 2.1    Related Work

Focusing on unsupervised similarity-based link prediction, we consider methods that calculate a bivariate function $\text{sim}(u, v) \in \mathbb{R}^d$, where $u, v \in V$ are nodes in a directed or undirected, possibly weighted graph $G = (V, E)$ [23, 24]. While similarity metrics often consider scalar functions ($d = 1$), recent vector space embeddings use binary operators to assign vector-valued "features" with $d > 1$ to node pairs. Since vectorial features are typically used in downstream classification techniques, this can be seen as an *implicit* mapping to similarities, for example "similar" features being assigned similar class probabilities. We limit our discussion to *topological or structural approaches* [23], and consider functions $\text{sim}(u, v)$ that can be calculated solely based on the edges $E$ in graph $G$ without requiring additional information such as node attributes or other non-topological graph properties.

Several works define scalar similarities based on local topological characteristics such as the Jaccard index of neighbour sets, degrees of nodes, or degree-weighted measures of common neighbours [25]. Other methods define similarities based on random walks, paths, or topological distance between nodes [9, 26–28]. Compared to purely local approaches, an advantage of random walk-based methods is their ability to incorporate both local and non-local information, which is crucial for sparse networks where nodes may lack common neighbours. Since walk-based methods reveal cluster patterns in networks [22], they generally perform well in downstream tasks such as link prediction and graph clustering [2]. Graph factorisation approaches that use eigenvectors of different types of *Laplacian matrices* that represent relationships between nodes share this high performance [29], likely because (i) Laplacians capture the dynamics of continuous-time random walks [30], and (ii) spectral methods can capture *small cuts* in graphs [31].

Building on these ideas, recent works on *graph representation learning* combine random walks and deep learning to obtain high-dimensional vector space embeddings of nodes, serving as features in downstream learning tasks [3, 14]: Perozzi et al. [6] generate a large number of short random walks to learn latent space representations of nodes by applying a word embedding technique that considers node sequences as word sequences in a sentence. This corresponds to an implicit factorisation of a matrix whose entries capture the logarithm of the expected probabilities to walk between nodes in a given number of steps [32]. Following a similar walk-based approach, Grover and Leskovec [7] generate node sequences with a biased random walker whose exploration behaviour can be tuned by *search bias* parameters $p$ and $q$. The resulting walk sequences are used as input for the word embedding algorithm `word2vec` [33], which embeds objects in a latent vector space with configurable dimensionality. Tang et al. [8] construct vector space embeddings of nodes that simultaneously preserve first- and second-order proximities between nodes. Similar to Adamic and Adar [25], second-order node proximities are defined based on common neighbours. Extending the

random walk approach in [6], Perozzi et al. [34] learn embeddings from so-called walklets, random walks that skip some nodes, resulting in embeddings that capture structural features at multiple scales.

The abovementioned graph embedding methods compute a representation of nodes in a, compared to the number of nodes in the network, low-dimensional Euclidean space. A suitably defined metric for *similarity* or *distance* of nodes enables recovering the link topology with high fidelity [35], forming the basis for similarity-based link prediction. In contrast, Lichtenwalter et al. [10] argued for a new perspective that uses supervised classifiers based on (i) multi-dimensional features of node pairs, and (ii) an undersampling of negative instances to address inherent class imbalances in link prediction. Recent applications of graph embedding to link prediction have taken a similar supervised approach, for example using vector-valued binary operators to construct features for node pairs from node vectors [6, 7, 24]. Despite good performance, recent works have cast a more critical light on such applications of low-dimensional graph embeddings. Questioning the distinction between deep learning-based embeddings and graph factorisation techniques, Qiu et al. [4] show that popular embedding techniques can be understood as (approximate) factorisations of matrices that capture graph topology. Thus, low-dimensional embeddings can be viewed as a (lossy) compression of graphs, while link prediction or graph reconstruction can be viewed as the decompression step. Fitting this view, a recent study of the topological characteristics of networks' low-dimensional Euclidean representations has highlighted fundamental limitations of embeddings to capture complex structures found in real networks [19].

Techniques like node2vec, LINE, or DeepWalk have been reported to perform well for link prediction despite those limitations. However, recent surveys concur that finetuning their hyperparameters to the specific data set is required [2, 21, 36], which can be problematic in large data sets and increase the risk of overfitting. When used for link prediction, graph embedding methods are typically combined with dimensionality reduction and supervised classification algorithms, possibly using non-linear kernels. Comparative studies found that the performance of Euclidean graph embeddings for link prediction is connected to their ability to represent communities in graphs as clusters in the feature space [2], which, due to the non-linear nature of graph data [37], strongly depends on their topology. Using symmetric operators or distance measures in metric spaces limits their ability to predict *directed* links because the ground truth for $(u, v)$ can differ from $(v, u)$ [15].

These issues raise the general question whether we should use low-dimensional Euclidean embeddings for link prediction tasks. Recent works addressed some of those open questions, for example with hyperbolic or non-linear embeddings [20, 37], extensions of Euclidean embeddings for directed link prediction [15], or embeddings that explicitly account for community structures [21, 38, 39]. However, existing works still use hyperparameters, require separate dimensionality reduction or model selection to identify the optimal number of dimensions, fail to capture rich hierarchically nested community structures present in real-world networks [40], or do not integrate community detection with representation learning. Addressing all issues at once, we take a novel approach that treats graph representation learning as a compression problem: We use the map equation [22], an analytical information-theoretic approach to compress flows of random walks in directed or undirected, possibly weighted networks based on their modular structure. Unlike recent work by Ghasemian et al. [41] that predicts links based on how they influence the map equation's estimated codelength, requiring inefficient recalculations, we take advantage of the map equation's coding machinery without any computational overhead. The map equation's hierarchical coding tree with node assignments provides an embedding in a discrete, non-metric latent space of possibly hierarchical community labels with automatically optimised dimensionality using a minimum description length approach. Following the map equation's compression principles, we relate the similarity between nodes $u$ and $v$ to how efficiently we can compress the link $(u, v)$ with respect to the network's modular structure. As an analytical approach, our method neither introduces hyperparameters nor needs to simulate random walks, and naturally yields asymmetric node similarities suitable to predict directed links.

## 2.2 Background: the map equation

The map equation is an information-theoretic objective function for community detection that, conceptually, models network flows with random walks [22]. To detect communities, the map equation compresses the random walks' per-step description length by searching for sets of nodes with long flow persistence: network areas where a random walker tends to stay for a longer time.

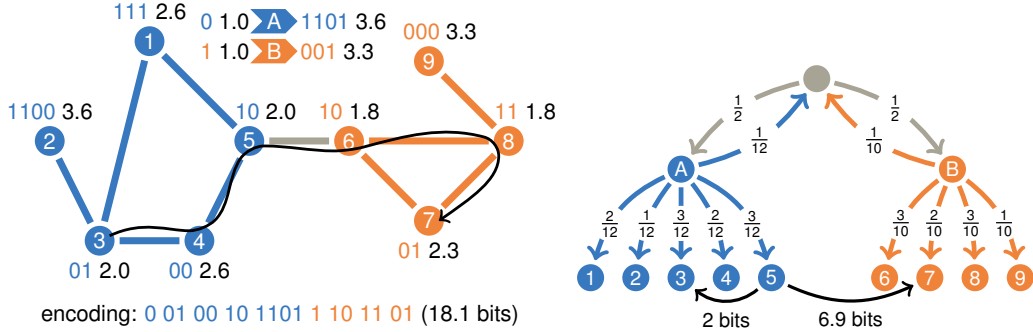

**Figure 2:** Map equation coding principles. **Left:** An example network with nine nodes, ten links, and two communities, A and B, indicated by colours. Each random-walker step is encoded by one codeword for intra-module transitions, or three codewords for inter-module transitions. Codewords are shown next to nodes in colours, their length in bits in the information-theoretic limit in black. Module entry and exit codewords are shown to the left and right of the coloured arrows, respectively. The black trace shows a possible section of a random walk with its encoding and theoretical length at the bottom. **Right:** The corresponding coding tree. Links are annotated with transition rates to calculate similarities in the information-theoretic limit. Each coding tree path corresponds to a network link, which may or may not exist. The coder remembers the random walker's module but not the most recently visited node. Describing the intra-module transition from node 5 to 3 requires $-\log_2\left(3/12\right) = 2$ bits. The inter-module transition from node 5 to 7 requires three steps and $-\log_2\left(1/12 \cdot 1/2 \cdot 2/10\right) \approx 6.9$ bits.

Consider a communication game where the sender observes a random walker on a network, and uses binary codewords to update the receiver about the random walker's location. In the simplest case, all nodes belong to the same module and we use a Huffman code to assign unique codewords to the nodes based on their stationary visit rates. With a one-module partition, $\mathsf{M}_1$, the sender communicates one codeword per random-walker step to the receiver. The theoretical lower limit for the per-step description length, we call it *codelength*, is the entropy of the nodes' visit rates [42],

$$L\left(\mathsf{M}_1\right) = \mathcal{H}\left(P\right) = -\sum_{u \in V} p_u \log_2 p_u, \tag{1}$$

where $\mathcal{H}$ is the Shannon entropy, $P$ is the set of the nodes' visit rates, and $p_u$ is node $u$'s visit rate.

In networks with modular structure, we can compress the random walks' description by grouping nodes into more than one module such that a random walker tends to remain within modules, and module switches become rare. This lets us re-use codewords across modules and design a codebook per module based on the nodes' module-normalised visit rates. However, sender and receiver need a way to encode module switches. The map equation uses a designated module-exit codeword per module and an index-level codebook with module-entry codewords. In a two-level partition, the sender communicates one codeword for intra-module random-walker steps to the receiver, or three codewords for inter-module steps (figure 2). The lower limit for the codelength is given by the sum of entropies associated with module and index codebooks, weighted by their usage rates. Given a partition of the network's nodes into modules, $\mathsf{M}$, the map equation [22] formalises this relationship,

$$L\left(\mathsf{M}\right) = q\mathcal{H}\left(Q\right) + \sum_{\mathsf{m} \in \mathsf{M}} p_\mathsf{m} \mathcal{H}\left(P_\mathsf{m}\right). \tag{2}$$

Here $q = \sum_{\mathsf{m} \in \mathsf{M}} q_\mathsf{m}$ is the index-level codebook usage rate, $q_\mathsf{m}$ is the entry rate for module $\mathsf{m}$, and $Q = \{q_\mathsf{m} \,|\, \mathsf{m} \in \mathsf{M}\}$ is the set of module entry rates; $\mathsf{m}_{\text{exit}}$ is the exit rate for module $\mathsf{m}$, $p_\mathsf{m} = \mathsf{m}_{\text{exit}} + \sum_{u \in \mathsf{m}} p_u$ is the codebook usage rate for module $\mathsf{m}$, and $P_\mathsf{m} = \{\mathsf{m}_{\text{exit}}\} \cup \{p_u \,|\, u \in \mathsf{m}\}$ is the set of node visit rates in $\mathsf{m}$, including $\mathsf{m}$'s module exit rate.

The map equation can detect communities in simple, weighted, directed, and higher-order networks, and can be generalised to hierarchical partitions through recursion [40]. To make use of node metadata for detecting communities, we can either incorporate a corresponding term in the map equation [43],

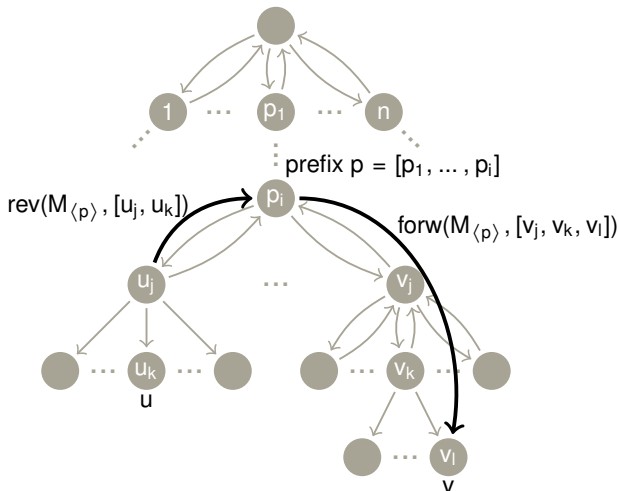

**Figure 3:** Illustration of map equation similarity between nodes $u$ and $v$ with addresses $\mathrm{addr}\,(\mathsf{M}, u) = [p_1, \ldots, p_i, u_j, u_k]$ and $\mathrm{addr}\,(\mathsf{M}, v) = [p_1, \ldots, p_i, v_j, v_k, v_l]$. $\mathsf{M}$ is the complete network partition. The longest common prefix between the addresses for $u$ and $v$ is $p = [p_1, \ldots, p_i]$, and $\mathsf{M}_{\langle p \rangle}$ is the sub-module at address $p$ within $\mathsf{M}$, that is the smallest module that contains $u$ and $v$.

design metadata-informed flow models [44], or introduce a prior network and reinforce link weights between nodes with the same metadata label [45].

## 3  MapSim: node similarities from modular flow compression

Compression-based similarity measures consider pairs of objects more similar if they jointly compress better. Extending this idea to networks, we exploit the coding of network flows based on the map equation, and use it to calculate information-theoretic pairwise similarities between nodes: *MapSim*. We interpret a network's community structure as an implicit embedding and, roughly speaking, consider nodes in the same community as more similar than nodes in different communities.

To calculate node similarities, we begin with a network partition and its corresponding modular coding scheme[2], which can be visualised as a tree, annotated with the transition rates defined by the link patterns in the network (figure 2). While the network's topology constrains random walks to transitions along existing links, the coding scheme is more flexible and can describe transitions between *any* pair of nodes. To describe the transition from node $u$ to $v$, we find the corresponding path in the partition tree and multiply the transition rates along that path, that is, we use the *coarse-grained description* of the network's community structure, not the network's actual link pattern; it can describe any transition regardless of whether the link $(u, v)$ exists in the network or not. The description length in bits for a path with transition rate $r$ is $-\log_2(r)$. For example, consider the scenario in figure 2 where we calculate similarity scores for the two directed links $(5, 3)$ and $(5, 7)$, neither of which exists in the network. Nodes 5 and 3 are in module $A$, and the rate at which a random walker in $A$ visits node 3 is $3/12$, requiring $-\log_2(3/12) = 2$ bits to describe that transition. Node 7 is in module $B$, and a random walker in $A$ exits $A$ at rate $1/12$, enters $B$ at rate $1/2$, and then visits node 7 at rate $2/10$, that is, at rate $1/120$, requiring $-\log_2(1/120) \approx 6.9$ bits.

Paths to derive similarities emanate from modules, not from nodes, because the model must generalise to unobserved data. If compression was our sole purpose, we would use node-specific codebooks containing codewords for neighbouring nodes, but no longer detect communities, and only be able to describe observed links. Instead, the map equation's coding scheme is designed to capitalise on modular network structures: The modular code structure provides a model that generalises to unobserved data, coarse-grains the path descriptions, and prevents overfitting.

For the general case, where $\mathsf{M}$ can be a hierarchical network partition, we number the sub-modules within each module $\mathsf{m}$ from 1 to $n_\mathsf{m}$ – we refer to these numbers as addresses – such that an ordered

---

[2]In principle, arbitrary network partitions can be used, regardless of the used community detection method.

sequence of addresses uniquely identifies a path starting at the root of the partition tree. We let $\mathrm{addr} \colon \mathsf{M} \times N \to List\,(\mathbb{N})$ be a function that takes a network partition and a node as input, and returns the node's address in the partition. To calculate the similarity of node $v$ to $u$, we identify the longest common prefix $p$ of the nodes' addresses, $\mathrm{addr}\,(\mathsf{M}, u)$ and $\mathrm{addr}\,(\mathsf{M}, v)$, and select the partition tree's sub-tree $\mathsf{M}_{\langle p \rangle}$ that corresponds to the prefix $p$: $\mathsf{M}_{\langle p \rangle}$ is the smallest sub-tree that contains $u$ and $v$. We obtain the addresses for $u$ and $v$ within sub-tree $\mathsf{M}_{\langle p \rangle}$ by removing the prefix $p$ from their addresses. That is, $\mathrm{addr}\,(\mathsf{M}, u) = p + \mathrm{addr}(\mathsf{M}_{\langle p \rangle}, u)$ and $\mathrm{addr}\,(\mathsf{M}, v) = p + \mathrm{addr}(\mathsf{M}_{\langle p \rangle}, v)$, where $+$ is list concatenation. The rate at which a random walker transitions from $u$ to $v$ is the product of (i) the rate at which the random walker moves along the path $\mathrm{addr}(\mathsf{M}_{\langle p \rangle}, u)$ in *reverse direction*, $\mathrm{rev}(\mathsf{M}_{\langle p \rangle}, \mathrm{addr}(\mathsf{M}_{\langle p \rangle}, u))$, that is from $u$ to the root of $M_{\langle p \rangle}$, and (ii) the rate at which the random walker moves along the path $\mathrm{addr}(\mathsf{M}_{\langle p \rangle}, v)$ in *forward direction*, $\mathrm{forw}(\mathsf{M}_{\langle p \rangle}, \mathrm{addr}(\mathsf{M}_{\langle p \rangle}, v))$, that is from the root of $\mathsf{M}_{\langle p \rangle}$ to $v$, where

$$\mathrm{rev}\,(\mathsf{M}, a) = \begin{cases} 1 & \text{if } a = [x] \\ \mathsf{M}_{\langle [x] \rangle, \mathrm{exit}} \cdot \mathrm{rev}(\mathsf{M}_{\langle [x] \rangle}, a') & \text{if } a = [x] + a' \end{cases} \tag{3}$$

$$\mathrm{forw}\,(\mathsf{M}, a) = \begin{cases} p_{\langle [x] \rangle} / p_{\mathsf{M}} & \text{if } a = [x] \\ \mathsf{M}_{\langle [x] \rangle, \mathrm{enter}} \cdot \mathrm{forw}(\mathsf{M}_{\langle [x] \rangle}, a') & \text{if } a = [x] + a' \end{cases} \tag{4}$$

and $a'$ denotes non-empty sequences. Here $p_{\mathsf{M}}$ is the codebook use rate for module $\mathsf{M}$ and $p_{\langle [x] \rangle}$ is the visit rate for the node identified by address $x$ within the given module. The final addresses in equation 3 and equation 4 are treated differently, reflecting that the map equation forgets the most recently visited node.

We illustrate these ideas in a generic example (figure 3). In short, we define map equation similarity,

$$\mathrm{MapSim}\,(M, u, v) = \mathrm{rev}(\mathsf{M}_{\langle p \rangle}, \mathrm{addr}(\mathsf{M}_{\langle p \rangle}, u)) \cdot \mathrm{forw}(\mathsf{M}_{\langle p \rangle}, \mathrm{addr}(\mathsf{M}_{\langle p \rangle}, v)), \tag{5}$$

where $p$ is the longest common prefix shared by the addresses of $u$ and $v$ in the partition tree defined by $\mathsf{M}$. To express map equation similarity in terms of description length, we take the $-\log_2$ of $\mathrm{MapSim}$ and regard pairs of nodes that yield a shorter description length as more similar.

$\mathrm{MapSim}$ is asymmetric since module entry and exit rates are, in general, different and $u$ and $v$ can have different visit rates. $\mathrm{MapSim}$ is zero if one node is in a disconnected component; the exit rate for regions without out-links is zero, so the corresponding description length is infinitely long. This issue can be addressed with the regularised map equation [45], a Bayesian approach that introduces an empirical prior to model incomplete data with weak links between all pairs of nodes, where prior link strengths depend on the connection patterns of each node.

We calculate node similarities in three steps: (i) inferring a network's community with Infomap [46], a greedy, search-based optimisation algorithm for the map equation, (ii) representing the corresponding coding scheme in a suitable data structure, and (iii) using *MapSim* to computing similarities based on the coding scheme. The overall approach is illustrated in figs. 1 – 3 and algorithm 1.

---

**Algorithm 1:** Pseudo-code of function $\mathrm{MapSim}$ to calculate similarity score for node pair $(u, v)$.

**Input** : graph $G$ and pair of nodes $(u, v)$
**Output**: similarity score of $(u, v)$

1 // Use Infomap to construct coding tree for compression
2 modules = `Infomap.minimiseMapEquation`$(G)$
3 tree = `buildPartitionTree`$(G, \mathsf{modules})$
4 p = `longestCommonPrefix`$(\mathsf{tree}, u, v)$
5 $\mathsf{tree}_{\langle p \rangle}$ = `smallestSubtree`$(\mathsf{tree}, \mathsf{p})$
6 // calculate code length of random walks from $u$ to $v$
7 addrU = `addr`$(\mathsf{tree}_{\langle p \rangle}, u)$
8 addrV = `addr`$(\mathsf{tree}_{\langle p \rangle}, v)$
9 revRate = `rev`$(\mathsf{tree}_{\langle p \rangle}, \mathsf{addrU})$
10 fwdRate = `forw`$(\mathsf{tree}_{\langle p \rangle}, \mathsf{addrV})$
11 **return** $-\log_2(\mathsf{revRate} \cdot \mathsf{fwdRate})$

---

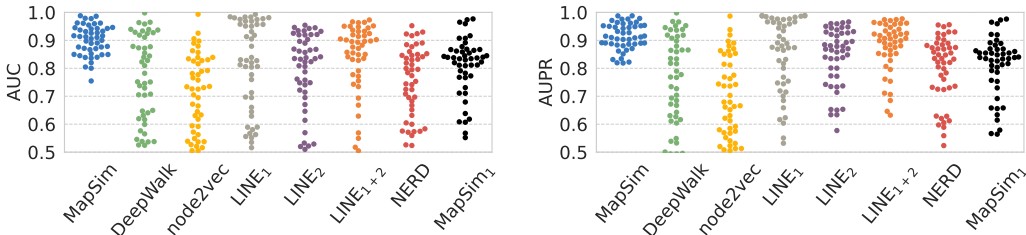

**Figure 4:** Link-prediction performance of MapSim, DeepWalk, node2vec, LINE, and NERD on 47 real-world networks. **Left:** AUC performance. **Right:** AUPR performance.

## 4    Experimental Validation

We evaluate the performance of *MapSim* in unsupervised, similarity-based link prediction for 47 real-world networks, 35 directed (table 1) and 12 undirected (table 3), retrieved from Netzschleuder [47] and Konect [48]. Details of the directed and undirected networks are shown in tables 2 and 4, respectively. Our analysis is based on a Python-implementation available on GitHub[3], building on Infomap, a fast and greedy search algorithm for minimising the map equation with an open source implementation in C++ [46, 49]. As baseline, we use four random walk and neighbourhood-based embedding methods: DeepWalk [6], node2vec [7], LINE [8], and NERD [15], using the respective author's implementation. We also include results for *MapSim* based on the one-module partition for each network for comparison, which ignores community structure. Adopting the argument by [7], we exclude graph factorisation methods and simple local similarity scores because they have already been shown to be inferior to node2vec. We include NERD because it is a recent random walk-based embedding method proposed for directed link prediction with higher reported performance than other walk-based embeddings [15].

### 4.1    Unsupervised Link Prediction

Different from works that use graph embeddings for *supervised* link prediction, we address *unsupervised* link prediction. Like Goyal and Ferrara [2] and Khosla et al. [15], we take a similarity-based approach that does not require training a classifier. We compute similarity scores based on node embeddings, rather than applying a supervised classifier to features computed for node pairs. We adopt the approach by Khosla et al. [15] and calculate node similarities as the sigmoid over the feature vectors' dot product.

Considering how different embedding techniques generalise across data sets, *we purposefully refrained from hyperparameter tuning*. We chose a single set of hyperparameters for each method, informed by the default parameters given by the respective authors and recent surveys' discussion regarding which hyperparameter values generally provide good link prediction performance. For DeepWalk and node2vec, we sample $r = 80$ random walks of length $l = 40$ per node, and use a window size of $w = 10$. For both methods, the underlying word embedding is applied using the default model parameters fixed by the authors, $skipgram = 1$, $k = 10$ and $mincount = 0$. For node2vec we set the return parameter to $p = 1$. Since for $q = p = 1$ node2vec is identical to DeepWalk, we use $q = 4$, which was found to provide good performance for link prediction [2]. We run LINE with first-order (LINE$_1$), second-order (LINE$_2$), and combined first-and-second-order proximity (LINE$_{1+2}$), use $1,000$ samples per node, and $s = 5$ negative samples. For NERD, we use $800$ samples and $\kappa = 3$ negative samples per node. We set the number of neighbourhood nodes to $n = 1$, as suggested by the authors for link prediction. We use $d = 128$ dimensions for all embeddings. Since *MapSim* is a non-parametric method, it does not require setting any hyperparameters. However, to avoid local optima when heuristically minimising the map equation, we run Infomap 100 times and select the partition with the shortest description length.

We use 5-fold cross-validation to split links into train and test sets, treating weighted links as indivisible. We calculate the node embedding (for *MapSim* the coding tree) in the training network, derive predictions based on node similarities, and evaluate them based on the links in the validation set.

---

[3]https://github.com/mapequation/map-equation-similarity

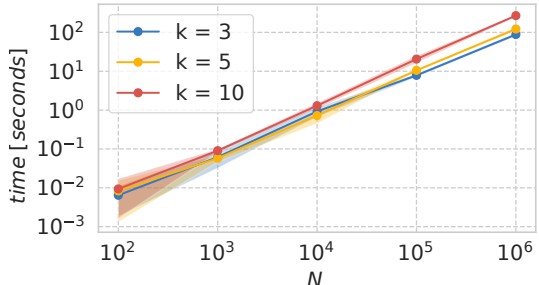

**Figure 5:** Runtime behavior for inferring the community structure with Infomap and constructing the coding tree for MapSim in synthetic $k$-regular networks with different size.

For each fold, we restrict the resulting training network to its largest (weakly) connected component. For a validation set with $k$ positive links, we sample $k$ negative links uniformly at random, and calculate scores for all $2k$ links. In undirected networks, for each positive link $(u, v)$, we also consider $(v, u)$ as positive, and, therefore, sample two negative links per positive link. Varying the discrimination threshold, we obtain a receiver operator characteristic (ROC) per fold, and calculate the area under the curve (AUC). Detailed results, including average and worst-case performance, are shown in tables 2 and 4; we also report precision-recall performance (table 5). We include *MapSim* based on the one-module partition[4] in the results and note that it performs better than using a modular partition in some cases: this suggests that the network does not have a strong community structure, which could be addressed with the regularised map equation [45]. When mentioning *MapSim* in the following, we refer to using modular partitions.

On average, *MapSim* outperforms all baseline methods across the 47 data sets in terms of AUC and AUPR (figure 4); for detailed results on a per-network basis see tables 2, 4, and 5 in the appendix. Using a one-sided two-sample $t$-test, we find that *MapSim*'s average performance across all networks is significantly higher than that of the best graph embedding method, LINE$_{1+2}$, both in directed and undirected networks ($p \approx 0.008$ and $p \approx 0.039$, respectively). *MapSim* provides the best performance in 11 of the 47 networks, with a standard deviation of the AUC score less than half of that of the best embedding-based method (LINE$_{1+2}$). For undirected networks, *MapSim* achieves the best performance for five of the 12 networks, while none of the embedding methods beats *MapSim's* performance in more than two networks. We find the largest performance gain in the directed network *linux*, where *MapSim* yields an increase of AUC of approximately 22.6% compared to the best embedding (NERD). *MapSim*'s worst-case performance across all networks is approximately 44% and 33% above that of the best-performing embedding for directed and undirected networks, respectively. *MapSim*' performance advantage can be as high as 84%, for example $AUC = 0.988$ of MapSim in *foursquare-friendships-new* vs. $AUC = 0.537$ for node2vec. While node2vec performs best in the largest directed network, *MapSim* performs best in the largest undirected network and in several small networks, suggesting that *MapSim* works well both for small and large networks.

We attribute those encouraging results to multiple features of our method: Different from graph embedding techniques that require downstream dimensionality reduction, *MapSim*'s compression approach implicitly includes model selection and avoids overfitting. Moreover, the representation of nodes in the coding tree is integrated with the optimisation of hierarchical community structures in the network. Due to its non-parametric approach and the use of the analytical map equation, *MapSim* performs well in absence of tuning to the specific data set.

### 4.2 Scalability Analysis

We analyse *MapSim*'s scalability in synthetically generated networks with modular structure and tunable size and link density. We generate $k$-regular random graphs with $N$ nodes and (mean) degree $k$. To avoid trivial configurations where a modular structure is absent, we create a network by first generating two $k$-regular random graphs with $\frac{N}{2}$ nodes each and "cross" two links, one from each of the two graphs, to obtain a single connected network with strong community structure. We then apply Infomap to (i) minimise the map equation and extract the network's modular structure, and

---

[4]With the one-module partition, MapSim becomes equivalent to preferential attachment.

(ii) construct the coding tree for calculating node similarities. We repeat this 10 times for random networks with different numbers of $N$ nodes and degrees $k$. The average run times are reported in figure 5, which shows that, for sparse networks, the runtime of *MapSim* is linear in the size of the network. Edler et al. [49] report that the theoretical asymptotic bound of computational complexity for the optimisation of the map equation is in $\mathcal{O}(NlogN)$, which is the same as for vector space embedding techniques like node2vec and DeepWalk[5]. Thus, *MapSim* does not entail higher computational complexity compared to popular graph embeddings. This makes it an interesting choice for practitioners looking for a simple and scalable method that works well in small, large, directed, and undirected networks.

## 5   Conclusion and Outlook

We propose *MapSim*, a novel information-theoretic approach to compute node similarities based on a modular compression of network flows. Different from vector space embeddings, *MapSim* represents nodes in a discrete, non-metric space of communities that yields *asymmetric* similarities suitable to predict links in *directed* and *undirected* networks. The results are highly interpretable because the network's modular structure explains the similarities. Using description length minimisation, *MapSim* naturally accounts for Occam's razor, which avoids overfitting and yields a parsimonious coding tree. Performing unsupervised link prediction, we compare *MapSim* to popular embedding-based algorithms on 47 data sets covering networks from a few hundred to hundreds of thousands of nodes and millions of edges. Our analysis shows that the average performance of *MapSim* is more than 7% higher than its closest competitor, outperforming all competing methods in 11 of the 47 networks. Taking a new perspective on graph representation learning, our work demonstrates the potential of compression-based methods with promising applications in other graph learning tasks. Moreover, recent generalisations of the map equation to temporal and higher-order networks [49] suggest that our method also applies to graphs with non-dyadic or time-stamped relationships.

## Acknowledgements

This work was partially supported by the Wallenberg AI, Autonomous Systems and Software Program (WASP) funded by the Knut and Alice Wallenberg Foundation. Ingo Scholtes acknowledges financial support from the Swiss National Science Foundation through grant No. 176938. Martin Rosvall was supported by the Swedish Research Council, Grant No. 2016-00796.

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

# A  Appendix

**Table 1:** Properties of 35 directed networks, where weighted networks are marked with W, temporal link counts before aggregation into a static network are marked with $*$, and $\rho$ is link reciprocity.

| Data | Ref | Nodes | Edges | $\rho$ |
|---|---|---|---|---|
| uni-email | [50] | 1,133 | 10,903 | 1.000 |
| polblogs | [51] | 1,490 | 19,090 | 0.243 |
| interactome-stelzl | [52] | 1,706 | 6,207 | 0.972 |
| interactome-figeys | [53] | 2,239 | 6,452 | 0.006 |
| us-air-traffic[W] | [54] | 2,278 | $*$6,390,340 | 0.757 |
| word-adjacency-japanese | [55] | 2,704 | 8,300 | 0.073 |
| openflights[W] | [56] | 3,214 | 66,771 | 0.978 |
| jdk | [48] | 6,434 | 150,985 | 0.009 |
| advogato[W] | [57] | 6,541 | 51,127 | 0.307 |
| word-adjacency-spanish | [55] | 11,586 | 45,129 | 0.091 |
| dblp-cite | [58] | 12,590 | 49,759 | 0.004 |
| anybeat | [59] | 12,645 | 67,053 | 0.535 |
| chicago-road | [60] | 12,982 | 39,018 | 0.943 |
| foldoc[W] | [61] | 13,356 | 120,238 | 0.479 |
| google | [62] | 15,763 | 171,206 | 0.254 |
| word-assoc[W] | [63] | 23,132 | 312,342 | 0.094 |
| cora | [64] | 23,166 | 91,500 | 0.051 |
| arxiv-citation-HepTh | [65] | 27,770 | 352,807 | 0.003 |
| digg-reply[W] | [66] | 30,398 | $*$87,627 | 0.002 |
| linux | [48] | 30,837 | 213,954 | 0.002 |
| arxiv-citation-HepPh | [65] | 34,546 | 421,578 | 0.003 |
| email-enron | [67] | 36,692 | 367,662 | 1.000 |
| inploid | [68] | 39,749 | 57,276 | 0.272 |
| pgp-strong | [69] | 39,796 | 301,498 | 0.660 |
| facebook-wall[W] | [70] | 46,952 | $*$876,993 | 0.588 |
| slashdot-threads[W] | [71] | 51,083 | $*$140,778 | 0.210 |
| python-dependency | [72] | 58,743 | 108,399 | 0.004 |
| lkml-reply[W] | [48] | 63,399 | $*$1,096,440 | 0.635 |
| epinions-trust | [73] | 75,888 | 508,837 | 0.405 |
| prosper | [48] | 89,269 | 3,394,979 | $< 0.001$ |
| google-plus | [74] | 211,187 | 1,506,896 | 0.482 |
| twitter-higgs-retweet[W] | [75] | 256,491 | 328,132 | 0.005 |
| amazon-copurchases-302 | [76] | 262,111 | 1,234,877 | 0.543 |
| notre-dame-web | [77] | 325,729 | 1,497,134 | 0.507 |
| twitter-followers | [78] | 465,017 | 834,797 | 0.003 |

**Table 2:** ROC AUC for link prediction in 35 directed networks for DeepWalk (DW), node2vec (n2v), LINE$_1$ (L$_1$), LINE$_2$ (L$_2$), LINE$_{1+2}$ (L$_{1+2}$), NERD, *MapSim* based on the one-level partition (MapSim$_1$), and *MapSim* based on modular partitions. Networks marked with W are weighted. † marks cases with AUC < 0.5 where we flipped the predicted link scores for AUC > 0.5. The best results per network are shown in bold, second-best underlined, and then rounded.

| Data | DW | n2v | L$_1$ | L$_2$ | L$_{1+2}$ | NERD | MS$_1$ | MS |
|---|---|---|---|---|---|---|---|---|
| uni-email | 0.911 | †0.505 | **0.957** | 0.903 | 0.932 | 0.667 | 0.711 | 0.852 |
| polblogs | 0.705 | 0.695 | 0.804 | 0.823 | 0.841 | 0.652 | 0.868 | **0.914** |
| interactome-stelzl | 0.810 | †0.505 | **0.913** | 0.758 | 0.849 | 0.524 | 0.710 | 0.755 |
| interactome-figeys | 0.524 | †0.828 | †**0.905** | 0.529 | †0.850 | 0.605 | 0.773 | 0.839 |
| us-air-traffic$^W$ | 0.649 | 0.572 | 0.563 | **0.935** | 0.933 | 0.774 | 0.858 | 0.916 |
| word-adjacency-japanese | †0.538 | †0.645 | †0.580 | 0.748 | 0.743 | 0.526 | **0.811** | 0.800 |
| openflights$^W$ | 0.782 | †0.665 | 0.918 | 0.934 | **0.948** | 0.708 | 0.838 | 0.941 |
| jdk | 0.746 | 0.857 | 0.820 | 0.695 | 0.755 | 0.725 | 0.974 | **0.986** |
| advogato$^W$ | 0.738 | 0.563 | 0.806 | 0.865 | **0.883** | 0.742 | 0.812 | 0.878 |
| word-adjacency-spanish | †0.538 | 0.672 | †0.713 | **0.824** | 0.791 | 0.632 | 0.811 | 0.805 |
| dblp-cite | 0.840 | †0.537 | †0.589 | 0.646 | 0.549 | 0.877 | 0.823 | **0.890** |
| anybeat | 0.647 | 0.539 | 0.644 | 0.841 | **0.857** | 0.683 | 0.834 | 0.850 |
| chicago-road | **0.998** | 0.816 | 0.981 | 0.670 | 0.835 | †0.583 | †0.608 | 0.848 |
| foldoc$^W$ | 0.927 | 0.549 | **0.951** | 0.832 | 0.905 | 0.571 | 0.618 | 0.845 |
| google | 0.844 | 0.792 | 0.831 | 0.868 | 0.896 | 0.697 | 0.867 | **0.962** |
| word-assoc$^W$ | 0.729 | 0.830 | 0.813 | 0.869 | **0.916** | 0.884 | 0.837 | 0.849 |
| cora | 0.939 | 0.839 | **0.950** | 0.761 | 0.831 | 0.830 | 0.839 | 0.906 |
| arxiv-citation-HepTh | 0.878 | 0.839 | **0.958** | 0.857 | 0.901 | 0.850 | 0.842 | 0.942 |
| digg-reply$^W$ | †0.546 | 0.618 | †0.552 | 0.714 | 0.693 | 0.841 | **0.845** | 0.836 |
| linux | 0.704 | 0.726 | 0.567 | 0.722 | 0.734 | 0.784 | 0.959 | **0.961** |
| arxiv-citation-HepPh | 0.959 | 0.897 | **0.975** | 0.835 | 0.898 | 0.860 | 0.830 | 0.942 |
| email-enron | 0.823 | †0.594 | **0.983** | 0.946 | 0.963 | 0.819 | 0.840 | 0.931 |
| inploid | 0.631 | 0.766 | 0.516 | 0.838 | 0.828 | 0.753 | 0.845 | **0.870** |
| pgp-strong | 0.873 | 0.527 | **0.984** | 0.890 | 0.924 | 0.795 | 0.782 | 0.925 |
| facebook-wall$^W$ | 0.877 | 0.789 | **0.931** | 0.809 | 0.855 | 0.813 | 0.768 | 0.867 |
| slashdot-threads$^W$ | 0.565 | 0.781 | 0.629 | 0.748 | 0.771 | 0.796 | **0.877** | 0.876 |
| python-dependency | 0.751 | 0.735 | †0.556 | 0.520 | †0.505 | 0.832 | **0.965** | 0.913 |
| lkml-reply$^W$ | 0.537 | 0.731 | 0.590 | **0.945** | 0.944 | 0.724 | 0.908 | 0.933 |
| epinions-trust | 0.599 | 0.777 | 0.806 | 0.943 | **0.952** | 0.887 | 0.916 | 0.937 |
| prosper | 0.828 | 0.631 | 0.697 | †0.614 | †0.518 | **0.952** | 0.891 | 0.945 |
| google-plus | 0.752 | 0.725 | **0.957** | 0.787 | 0.893 | 0.891 | 0.862 | 0.946 |
| twitter-higgs-retweet$^W$ | 0.620 | 0.879 | †0.695 | †0.522 | †0.569 | 0.799 | **0.977** | 0.820 |
| amazon-copurchases-302 | 0.963 | 0.826 | **0.980** | 0.896 | 0.936 | 0.575 | 0.638 | 0.910 |
| notre-dame-web | 0.965 | 0.926 | **0.975** | 0.919 | 0.964 | 0.923 | 0.867 | 0.962 |
| twitter-followers | 0.526 | †**0.993** | †0.993 | 0.510 | †0.973 | 0.917 | 0.809 | 0.871 |
| Average | 0.750 | 0.719 | 0.802 | 0.786 | 0.832 | 0.757 | 0.830 | **0.892** |
| Worst | 0.524 | 0.505 | 0.516 | 0.510 | 0.505 | 0.524 | 0.608 | **0.755** |
| Standard Deviation | 0.148 | 0.131 | 0.164 | 0.129 | 0.128 | 0.118 | 0.088 | **0.054** |

**Table 3:** Properties of 12 undirected networks, where weighted networks are marked with W.

| Data | Ref | Nodes | Edges |
|------|-----|-------|-------|
| new-zealand-collab[W] | [79] | 1,511 | 4,273 |
| urban-streets-venice | [80] | 1,840 | 2,407 |
| urban-streets-ahmedabad | [80] | 2,870 | 4,387 |
| power | [81] | 4,941 | 6,594 |
| facebook-organizations-L1 | [82] | 5,793 | 45,266 |
| reactome | [83] | 6,327 | 147,547 |
| physics-collab-arXiv[W] | [84] | 14,488 | 59,026 |
| marvel-universe | [85] | 19,428 | 95,497 |
| internet-as | [86] | 22,963 | 48,436 |
| marker-cafe | [74] | 69,413 | 1,644,849 |
| livemocha | [48] | 104,103 | 2,193,083 |
| foursquare-friendships-new | [87] | 114,324 | 607,333 |

**Table 4:** ROC AUC on 12 undirected networks for DeepWalk (DW), node2vec (n2v), LINE$_1$ (L$_1$), LINE$_2$ (L$_2$), LINE$_{1+2}$ (L$_{1+2}$), NERD, *MapSim* based on the one-level partition (MS$_1$), and *MapSim* based on modular partitions (MS). Networks marked with W are weighted. † marks cases with AUC $< 0.5$ where we flipped the predicted link scores for AUC $> 0.5$. The best results per network are shown in bold, second-best underlined, and then rounded.

| Data | DW | n2v | L$_1$ | L$_2$ | L$_{1+2}$ | NERD | MS$_1$ | MS |
|------|-----|-----|-----|-----|-----|------|------|-----|
| new-zealand-collab[W] | 0.616 | 0.734 | †0.660 | **0.921** | 0.895 | †0.559 | 0.834 | 0.839 |
| urban-streets-venice | 0.872 | 0.834 | 0.777 | 0.570 | 0.668 | 0.573 | †0.607 | **0.889** |
| urban-streets-ahmedabad | **0.939** | 0.890 | 0.828 | †0.533 | 0.629 | †0.575 | †0.731 | 0.897 |
| power | 0.919 | 0.863 | 0.827 | 0.741 | 0.777 | 0.600 | 0.552 | **0.959** |
| facebook-organizations-L1 | 0.937 | 0.516 | 0.968 | 0.954 | 0.966 | 0.846 | 0.864 | **0.979** |
| reactome | 0.934 | 0.592 | **0.983** | 0.925 | 0.950 | 0.846 | 0.820 | 0.978 |
| physics-collab-arXiv[W] | 0.929 | 0.521 | **0.977** | 0.807 | 0.871 | 0.695 | 0.568 | 0.955 |
| marvel-universe | 0.854 | †0.633 | 0.879 | 0.834 | **0.902** | 0.852 | 0.679 | 0.900 |
| internet-as | 0.641 | †0.705 | 0.535 | 0.921 | 0.920 | 0.744 | 0.766 | **0.927** |
| marker-cafe | 0.576 | 0.906 | 0.760 | 0.920 | 0.914 | **0.930** | 0.907 | 0.916 |
| livemocha | 0.708 | 0.758 | 0.839 | 0.861 | 0.876 | **0.924** | 0.855 | 0.876 |
| foursquare-friendships-new | 0.924 | 0.537 | 0.968 | 0.932 | 0.950 | 0.836 | 0.791 | **0.988** |
| Average | 0.821 | 0.707 | 0.834 | 0.826 | 0.860 | 0.748 | 0.748 | **0.925** |
| Worst | 0.576 | 0.521 | 0.535 | 0.533 | 0.629 | 0.559 | 0.552 | **0.839** |
| Standard Deviation | 0.136 | 0.140 | 0.132 | 0.137 | 0.106 | 0.136 | 0.116 | **0.045** |

**Table 5:** Average precision on 47 directed and undirected networks for DeepWalk (DW), node2vec (n2v), $LINE_1$ ($L_1$), $LINE_2$ ($L_2$), $LINE_{1+2}$ ($L_{1+2}$), NERD, *MapSim* based on the one-level partition ($MapSim_1$), and *MapSim* based on modular partitions. Weighted networks are marked with W. Results marked with † correspond to cases with AUC $< 0.5$ where we flipped the predicted link scores. Results are rounded, the best results are shown in bold, second-best are underlined.

| Data | DW | n2v | $L_1$ | $L_2$ | $L_{1+2}$ | NERD | $MS_1$ | MS |
|---|---|---|---|---|---|---|---|---|
| uni-email | 0.914 | †0.513 | **0.964** | 0.916 | 0.940 | 0.736 | 0.692 | 0.870 |
| polblogs | 0.627 | 0.631 | 0.817 | 0.838 | 0.853 | 0.724 | 0.851 | **0.903** |
| new-zealand-collab$^W$ | 0.643 | 0.661 | †0.768 | **0.925** | 0.907 | †0.606 | 0.855 | 0.865 |
| interactome-stelzl | 0.835 | †0.513 | **0.944** | 0.773 | 0.853 | 0.612 | 0.757 | 0.820 |
| urban-streets-venice | **0.897** | 0.870 | 0.828 | 0.634 | 0.711 | 0.597 | †0.564 | 0.890 |
| interactome-figeys | 0.533 | †0.703 | †**0.889** | 0.653 | †0.865 | 0.730 | 0.730 | 0.819 |
| us-air-traffic$^W$ | 0.616 | 0.552 | 0.685 | **0.937** | 0.934 | 0.835 | 0.833 | 0.903 |
| word-adjacency-japanese | †0.494 | †0.570 | †0.623 | 0.801 | 0.796 | 0.628 | **0.855** | 0.831 |
| urban-streets-ahmedabad | **0.953** | 0.919 | 0.864 | †0.577 | 0.685 | †0.523 | †0.658 | 0.915 |
| openflights$^W$ | 0.767 | †0.621 | 0.934 | 0.950 | **0.960** | 0.798 | 0.840 | 0.950 |
| power | 0.936 | 0.897 | 0.874 | 0.800 | 0.828 | 0.620 | 0.566 | **0.962** |
| facebook-organizations-L1 | 0.919 | 0.508 | **0.977** | 0.966 | 0.974 | 0.882 | 0.835 | 0.976 |
| reactome | 0.908 | 0.580 | **0.985** | 0.944 | 0.961 | 0.890 | 0.786 | 0.978 |
| jdk | 0.777 | 0.862 | 0.891 | 0.737 | 0.807 | 0.761 | 0.973 | **0.987** |
| advogato$^W$ | 0.769 | 0.505 | 0.868 | 0.892 | **0.905** | 0.805 | 0.810 | 0.890 |
| word-adjacency-spanish | †0.496 | 0.652 | †0.754 | **0.863** | 0.848 | 0.732 | 0.863 | 0.851 |
| dblp-cite | 0.834 | †0.485 | †0.551 | 0.742 | 0.646 | **0.908** | 0.828 | 0.905 |
| anybeat | 0.672 | 0.523 | 0.748 | 0.884 | **0.894** | 0.784 | 0.867 | 0.883 |
| chicago-road | **0.998** | 0.863 | 0.986 | 0.735 | 0.874 | †0.559 | †0.579 | 0.909 |
| foldoc$^W$ | 0.946 | 0.575 | **0.966** | 0.848 | 0.914 | 0.629 | 0.658 | 0.888 |
| physics-collab-arXiv$^W$ | 0.939 | 0.592 | **0.983** | 0.858 | 0.899 | 0.725 | 0.634 | 0.964 |
| google | 0.859 | 0.775 | 0.903 | 0.878 | 0.907 | 0.775 | 0.889 | **0.976** |
| marvel-universe | 0.864 | †0.666 | **0.914** | 0.840 | 0.899 | 0.884 | 0.615 | 0.910 |
| internet-as | 0.685 | †0.742 | 0.659 | 0.930 | 0.930 | 0.822 | 0.817 | **0.932** |
| word-assoc$^W$ | 0.727 | 0.846 | 0.873 | 0.896 | **0.922** | 0.902 | 0.848 | 0.862 |
| cora | 0.938 | 0.815 | **0.958** | 0.834 | 0.880 | 0.847 | 0.826 | 0.926 |
| arxiv-citation-HepTh | 0.865 | 0.812 | **0.966** | 0.896 | 0.925 | 0.868 | 0.839 | 0.952 |
| digg-reply$^W$ | †0.501 | 0.585 | †0.604 | 0.772 | 0.761 | **0.873** | 0.835 | 0.834 |
| linux | 0.734 | 0.663 | 0.701 | 0.733 | 0.754 | 0.835 | 0.959 | **0.965** |
| arxiv-citation-HepPh | 0.952 | 0.890 | **0.975** | 0.881 | 0.923 | 0.870 | 0.813 | 0.952 |
| email-enron | 0.816 | †0.541 | **0.988** | 0.963 | 0.974 | 0.873 | 0.860 | 0.949 |
| inploid | 0.667 | 0.736 | 0.532 | 0.879 | 0.875 | 0.819 | 0.869 | **0.891** |
| pgp-strong | 0.879 | 0.568 | **0.989** | 0.927 | 0.946 | 0.848 | 0.804 | 0.954 |
| facebook-wall$^W$ | 0.865 | 0.744 | **0.951** | 0.865 | 0.890 | 0.833 | 0.753 | 0.890 |
| slashdot-threads$^W$ | 0.604 | 0.769 | 0.744 | 0.835 | 0.848 | 0.855 | 0.883 | **0.886** |
| python-dependency | 0.790 | 0.763 | †0.715 | 0.653 | †0.632 | 0.889 | **0.965** | 0.915 |
| lkml-reply$^W$ | 0.494 | 0.612 | 0.719 | **0.959** | 0.958 | 0.821 | 0.920 | 0.942 |
| marker-cafe | 0.539 | 0.853 | 0.832 | 0.921 | 0.917 | **0.949** | 0.901 | 0.912 |
| epinions-trust | 0.611 | 0.679 | 0.875 | 0.960 | **0.964** | 0.925 | 0.921 | 0.947 |
| prosper | 0.818 | 0.531 | 0.616 | †0.650 | †0.465 | **0.956** | 0.855 | 0.927 |
| livemocha | 0.694 | 0.737 | 0.880 | 0.868 | 0.884 | **0.930** | 0.854 | 0.881 |
| foursquare-friendships-new | 0.918 | 0.520 | 0.976 | 0.948 | 0.961 | 0.858 | 0.792 | **0.988** |
| google-plus | 0.704 | 0.679 | **0.960** | 0.870 | 0.921 | 0.921 | 0.870 | 0.960 |
| twitter-higgs-retweet$^W$ | 0.630 | 0.880 | †0.800 | †0.634 | †0.707 | 0.874 | **0.976** | 0.822 |
| amazon-copurchases-302 | 0.966 | 0.850 | **0.987** | 0.931 | 0.957 | 0.589 | 0.656 | 0.946 |
| notre-dame-web | 0.967 | 0.938 | **0.980** | 0.946 | 0.971 | 0.930 | 0.891 | 0.971 |
| twitter-followers | 0.549 | †0.987 | †**0.989** | 0.714 | †0.977 | 0.955 | 0.839 | 0.887 |

