# OpenReview forum: "Similarity-based Link Prediction from Modular Compression of Network Flows"
_logconference.io/LOG/2022/Conference — LoG 2022 Poster_

### Official Review · Reviewer_TcAc · 2022-10-17

**Overall Score:** 5
**Confidence:** 3

**Review:**

This paper presents an information-theoretic framework, MapSim, to compute node similarities based on a modular compression of network flows. Specifically, The framework first infers a given graph’s hierarchical community structure using InfoMap, an optimization algorithm for the map equation, and represents the corresponding coding scheme. Then MapSim computes similarities between nodes based on the coding scheme. Experiments on 47 real-world graph datasets show the effectiveness of MapSim for link prediction compared to random walk-based embedding methods.

Strengths
- The idea of formulating graph representation learning as a compression problem and applying the map equation for link prediction is interesting and novel.
- Experiments were conducted on various graph datasets and showed competitive performances compared to random walk-based embedding methods.

Weakness
- My main concern is the lack of comparison with GNN-based methods. Many recent papers on link prediction have already shown that GNN-based methods outperform random walk-based embedding methods. In particular, like the proposed method, GNN-based methods [1,2,3] designed to use only structural features have been proposed and outperformed the random-walk based embedding methods. Therefore, In terms of graph representation for link prediction, comparison with GNN-based methods is necessary.

- The proposed framework is limited to being designed to utilize only structural features. Even though using only structural features has the advantage that it can be applied to any graph dataset, there can be cases where input features given to each node are meaningful for link prediction.

[1] Zhang et al., "Link prediction based on graph neural networks." Advances in neural information processing systems (2018).

[2] Yun et al., "Neo-gnns: Neighborhood overlap-aware graph neural networks for link prediction." Advances in Neural Information Processing Systems (2021)

[3] Zhaocheng et al., “Neural Bellman-Ford Networks: A General Graph Neural Network Framework for Link Prediction.” Advances in Neural Information Processing Systems (2021)

---

### Official Review · Reviewer_4x6E · 2022-10-22

**Overall Score:** 6
**Confidence:** 3

**Review:**

Summary:

The paper introduces MapSim, a method to estimate node similarities based on the Map Equation.  The method has several advantages including an ability to compute asymmetric similarities for directed graphs and an average performance greater than competing methods such as node2vec.  The method is novel as far as I am aware yet the insight to describe distances between nodes using the coding scheme from the map equation seems natural.

I believe the paper is appropriate for the conference as it demonstrates the utility of information-theoretic measures to the important task of node similarity computation.

Recommendation: Accept.

Strong Points:
* The methods are compared across 47 datasets including 35 directed and 12 directed graphs using different splits of the data.  The performance improvement is statistically compared to that of competing methods and is a significant improvement.
* The method performs well compared to competing methods and scalability analysis is run to demonstrate that the runtime is roughly linear in the size of the network (and upper bounded by N logN where N is the number of nodes in the network).

Weak Points:
* The authors mention that they purposely refrain from hyperparameter tuning which may disadvantage some of the competing methods.
* The method is heavily reliant on the Map Equation and Infomap and the theoretical novelty is therefore limited to the method for computing node similarities.


Questions:
1. How would the performance comparison change with hyperparameter tuning?

---

### Official Review · Reviewer_DTkm · 2022-10-23

**Overall Score:** 6
**Confidence:** 5

**Review:**

### Overview

The paper proposes a new methodology for node similarity computation on graphs. The proposed metric, MapSim, is based on information-theoretic and compression principles. In particular, MapSim leverages the idea of modular compression of network flows, as it has been proposed by the seminal InfoMap algorithm for community detection. In the context of node similarity estimation, MapSim constitutes an unsupervised learning framework where node embeddings are derived from the position of nodes in the coding tree, and their corresponding similarities are computed based on how well random walks are compressed.  The paper presents in detail the formulation of the methodology using toy examples whenever this is necessary. The proposed methodology is evaluated on the link prediction task on several real-world datasets.

**Strengths:**
- The formulation of the objective function is very interesting.
- Experiments have been conducted on a large number of datasets.

**Limitations:**
-  The main limitation of the paper has to do with the choice of baseline models. A more detailed discussion follows below.
- Although the paper is clearly written, I would expect to have some of the experiments shown in the main paper and not in the Appendix. Therefore, some restructuring of the paper might be necessary.

### Detailed comments

* The basic idea of the paper follows the methodology presented in the InfoMap algorithm for community detection. This slightly limits the novelty of the proposed model.

* The empirical analysis of the proposed methodology has been based on a large number of real-world datasets. This definitely provides a strong evidence about the performance of the model. Nevertheless, I have some concerns about the choice of baseline models.
   - First, despite all corresponding to seminal papers, there exist more recent models which outperform DeepWalk, node2vec, and LINE. Why more recent baselines, haven’t been used (in addition to DeepWalk, node2vec, and LINE?
  - Second, none of the used baselines takes into account the community structure of the graph. This basic assumption of MapSim is that the underlying graphs have some modular structure. There are recent baselines which incorporate information about the community structure in the computation of the embeddings.
  - Third, the experiments have been conducted only for the link prediction task. Is there any particular reason for that? Even if node classification is based on different principles, the paper could examine the performance of MapSim in node clustering.

* Lastly, I have a general comment about the structure of the paper. It would be interesting to give a sample of the link prediction experiments in the main paper. In its current form, all the key experiments are given in the appendix.

---

### Meta-Review · Area_Chair_xsvK · 2022-11-13

**Confidence:** 3
**Recommendation:** Accept

**Meta Review:**

The paper proposes a new methodology, MapSim, for node similarity computation on graphs, based on information-theoretic and compression principles. Reviewers have reached a consensus on the fact that the methodology is interesting and novel, and the experimental results are competitive. The main concerns are the organization of the paper (most experiments are shown in the main paper and not in the Appendix), and the lack of more recent baseline methods and GNN-based link prediction baselines.

Three critical strong/weak points:

S1: The proposed methodology is interesting and novel.

W1: The lack of more recent baseline methods and GNN-based link prediction baselines.

W2: The organization of the paper

where S1>W1>W2

Overall, I would highly recommend the authors stress the suggestion mentioned above to improve the paper. Despite there being room for improving the paper, I hope the creativity of the novelty research idea can be acknowledged as well. Therefore, I recommend the acceptance of the paper.

---

### Decision · Program_Chairs · 2022-11-23

Accept (Poster)